# Characterization of the Complete Mitochondrial Genome of *Agelas nakamurai* from the South China Sea

**DOI:** 10.3390/ijms25010357

**Published:** 2023-12-26

**Authors:** Zijian Lu, Qiang Lin, Huixian Zhang

**Affiliations:** 1CAS Key Laboratory of Tropical Marine Bio-Resources and Ecology (LMB), Guangdong Provincial Key Laboratory of Applied Marine Biology, South China Sea Institute of Oceanology, Chinese Academy of Sciences, Guangzhou 510000, China; luzijian22@mails.ucas.ac.cn; 2Southern Marine Science and Engineering Guangdong Laboratory (Guangzhou), Guangzhou 510000, China; 3University of Chinese Academy of Sciences, Beijing 100000, China

**Keywords:** *Agelas*, mitochondrial genome, phylogeny, positive selection

## Abstract

The *Agelas* genus sponges are widely distributed and provide shelter for organisms that inhabit reefs. However, there is a lack of research on the genetic diversity of the *Agelas* sponges. Additionally, only one *Agelas* mitochondrial genome has been documented, leaving the characteristics of the *Agelas* genus’s mitogenome in need of further clarification. To address this research gap, we utilized Illumina HiSeq4000 sequencing and de novo assembly to ascertain the complete mitochondrial genome of *Agelas* sp. specimens, sourced from the South China Sea. Our analysis of the *cox1* barcoding similarity and phylogenetic relationship reveals that taxonomically, the *Agelas* sp. corresponds to *Agelas nakamurai*. The mitogenome of *Agelas nakamurai* is 20,885 bp in length, encoding 14 protein-coding genes, 24 transfer RNA genes, and 2 ribosomal RNA genes. Through a comparison of the mitochondrial genes, we discovered that both *Agelas nakamurai* and *Agelas schmidti* have an identical gene arrangement. Furthermore, we observed a deletion in the *trnD* gene and duplication and remodeling of the *trnL* gene in the *Agelas nakamurai*’s mitogenome. Our evolutionary analysis also identified lineage-specific positive selection sites in the *nad3* and *nad5* genes of the *Agelas* sponges’ mitogenome. These findings shed light on the gene rearrangement events and positive selection sites in the mitogenome of *Agelas nakamurai*, providing valuable molecular insights into the evolutionary processes of this genus.

## 1. Introduction

*Agelas* sponges are notable reef organisms due to their large size and vibrant colors [1]. They are widely distributed across the tropical western Atlantic, the temperate north Atlantic (including the Mediterranean), and the central and western Indo-Pacific seas, inhabiting depths ranging from 2 to 400 m [2]. The ecological roles of *Agelas* sponges include providing habitat and shelter for various marine organisms. Their complex body structure creates crevices and spaces that serve as hiding places for small fish, crustaceans, and other invertebrates seeking refuge from predators. Additionally, these sponges offer a surface for attachment to diverse organisms such as algae, polychaetes, and bryozoans [3,4,5]. Moreover, *Agelas* sponges possess a substantial amount of biologically active secondary metabolites, comprising alkaloids (particularly bromopyrrole derivatives), terpenoids, sphingolipids, carotenoids, and fatty acids [6]. Consequently, they have emerged as a crucial source of novel pharmaceutical compounds.

The mitochondrial genome is commonly employed in species classification [7], population genetics [8], molecular ecology [9], and molecular phylogeography [10] because of its abundance of rare genomic features that are highly valuable for constructing phylogenetic trees. These features include indels in coding sequences, variations in genetic codes, alterations in the secondary structure of tRNAs and rRNAs, as well as gene rearrangements [11]. Sponges possess a more diverse mitogenome compared with bilaterian animals [12]. Previous studies have uncovered intriguing phenomena observed in multiple sponge mitogenomes, including the loss of tRNA genes, the presence of repetitive elements with hairpin structures [13], and the phenomenon of multipartite mtDNAs, like those found in calcareous sponges [14,15,16,17]. However, these findings are limited to specific genera or species with sequenced mitogenomes, leaving numerous unexplored molecular evolutionary phenomena in sponges awaiting investigation. Although there has been a substantial increase in sponge mitogenome sequencing in the past decade [15,18,19], there is still a lack of data on *Agelas* mitogenomes. To date, only one species from the *Agelas* genus, *Agelas schmidti* (Accession: NC_010213.1) [20], has undergone mitochondrial genome sequencing. This dearth of information impedes our comprehensive understanding of the mitogenome details of *Agelas* sponges.

Here, we present the first report on the complete mitogenome of an *Agelas* sp. discovered in the South China Sea. The objectives of our study were: (1) to identify *Agelas* sp. at the species level; (2) to reconstruct the phylogenetic relationships among Demospongiae sponges using mitochondrial protein-coding genes (PCGs); and (3) to investigate the linear characteristics and adaptive evolution of genes in the mitogenomes of *Agelas*. Overall, our study aimed to provide a valuable molecular reference for future research on the ecological and evolutionary aspects of sponge mitogenomes.

## 2. Results

### 2.1. Sample Collection, Sequencing Quality and Assembly Results

Sponge specimens were collected at Quanfu Island of Xisha Islands from the South China sea. Through HiSeq 4000 platform sequencing, a total of 2,586,488,700 bp of clean reads were obtained after removing adapter sequences and low-quality sequences. The base quality Q20 value was 96.39%, the Q30 value was 91.31% and the GC content was 50.97%. The Chloroplast & Mitochondrial Assembly (CMA) V1.1.1 software (Guangzhou SCGene Co., Ltd., Guangzhou, China) was used for genome assembly. Based on the assembled complete mitochondrial genome sequence, the captured mitochondrial genome sequencing reads were statistically analyzed using DNA sequence alignment. The genome depth was calculated to be 68.13×, and the genome coverage was 100%.

### 2.2. Species Identification

Based on the morphological characteristics (Appendix A), the specimens were tentatively identified as *Agelas* in taxonomy. To achieve species-level identification, a Maximum Likelihood (ML) phylogenetic tree was constructed using the *cox1* gene as the barcode sequence (Figure 1). In the ML tree, species belonging to the *Agelas* were grouped into distinct clades, with strong support from high bootstrap values, suggesting that the phylogenetic analysis based on the *cox1* gene accurately classifies different *Agelas* species. Importantly, the individual we investigated clusters together with two other *Agelas nakamurai* specimens (Accession: DQ069305.1, DQ069304.1) within the same clade. Moreover, their *cox1* barcoding showed complete similarity of 100% (query coverage: 63.3%), providing robust evidence for the taxonomic classification of the studied individual as *Agelas nakamurai* at the species level. Consequently, “*Agelas nakamurai*” would be used instead of “*Agelas* sp.” in the subsequent text.

### 2.3. Mitogenome Composition

The *Agelas nakamurai* mitogenome has a total length of 20,885 bp (GenBank accession No. OQ363829), consisting of 27.7% A bases, 34.7% T bases,15.9% C bases, and 21.7% G bases, resulting in a total GC content of 37.6%. The mitogenome contains a total of 40 genes: 2 rRNA genes (*rnl* & *rns*), 14 PCGs (*nad1-6, 4l*; *cox1-3*; *atp6*, *8*, *9* and *cob*), 24 tRNA genes (*trnR*_1_, *Q*, *W*, *T*, *S*_1_, *K*, *N*, *M*_1_, *H*, *P*, *E*, *R*_2_, *S*_2_, *A*, *C*, *L*_1_, *L*_2_, *Y*, *M*_2_, *M*_3_, *I*, *F*, *G*, *V*) (Figure 2). Among the 14 PCGs, the start codons for *nad6* and *atp6* are GTG, while the remaining coding genes initiate with ATG. The stop codons of *nad1*, *nad3*, and *cob* are TAG, while the rest of the PCGs end with TAA (Appendix A). No introns were found in any of the PCGs.

The mitogenome of *Agelas nakamurai* exhibits a highly compact structure, with coding sequences making up 86.7% of the genome, while the non-coding region makes up 13.3%. This compactness is primarily due to overlapping genes and the tight arrangement of adjacent genes. For instance, the *trnE* gene overlaps with the *nad6* gene by 3 base pairs, and the *trnR* gene is closely connected to the *nad4l* gene without any intervening spacer sequence.

### 2.4. The Phylogenetic Relationship and Gene Arrangement

A multigene phylogenetic tree was constructed for 38 Demospongiae sponges by utilizing PCGs obtained from mitogenomes (Appendix A). Bayesian inference (BI) and ML analyses produced the same phylogenetic topology (Figure 2; Appendix A), and most of their internal nodes are well supported. The multigene phylogenetic tree (Figure 3) reveals that the *Agelas nakamurai* and *Agelas schmidti* are clustered together and share a close affinity with *Axinella corrugata*.

Gene arrangement diagrams reveal that the number of PCGs and rRNA genes in Demospongiae sponges is consistently maintained, with a highly conserved order of arrangement. The majority of PCGs and rRNA genes in sponges are organized in the following sequence: *cox*, *nad1*, *nad2*, *nad5*, *rnS*, *rnL*, *cox2*, *atp8*, *atp6*, *cox3*, *cytb*, *atp9*, *nad4*, *nad6*, *nad3*, and *nad4L*. However, certain lineages present rearrangements, particularly at the superfamily level, in which Dictyoceratida and Dendroceratida display unique arrangements of PCGs and rRNA genes. Additionally, some species, such as *Topsentia ophiraphidites* and *Axinella corrugata*, that cannot be accurately clustered into the same clade as their superfamily, exhibit distinct orders of PCGs and rRNA arrangements. On the contrary, tRNA genes within Demospongiae display significant variation in both quantity and arrangement. Loss and rearrangement of tRNA genes are common phenomena.

The visualization of mitochondrial genomes in the genus *Agelas* (Appendix A) indicates a disparity in genome size between *Agelas nakamurai* and *Agelas schmidti*, primarily attributed to non-coding sequences. Both species exhibit identical gene arrangements, including PCGs, rRNAs, and tRNAs. This finding suggests a potential ancestral characteristic within the *Agelas*. Based on the detailed comparison with *Axinella corrugata* (Appendix A) and the gene arrangement information of other Demospongiae sponges (Figure 3), we observe that the gene arrangement of the *Agelas* sponge mitogenome exhibits lineage specificity primarily in the number and arrangement order of tRNAs, although its PCGs and rRNA genes align with most sponges. Interestingly, we found that neither *Agelas nakamurai* nor *Agelas schmidti* contain the *trnD* gene, which is uncommon among Demospongiae sponges. In comparison, all other sponge species, except for those in the Dictyoceratida order which have experienced significant loss of tRNA genes, possess the *trnD* gene.

### 2.5. tRNA Phylogenetic Analysis and Similarity Comparison

To examine potential instances of tRNA duplication and remodeling in the mitochondrial genome of *Agelas nakamurai*, we extracted its tRNA genes and constructed phylogenetic trees together with the tRNA genes of *Ectyoplasia ferox*, *Geodia neptuni*, *Halichondria okadai*, *Agelas schmidti*, and *Axinella corrugata*. Surprisingly, we discovered that the phylogenetic relationship between the *trnL*_1_*(uaa)* and *trnL*_2_*(uag)* genes of *Agelas nakamurai* differ significantly from that of other sponges. The phylogenetic analysis of tRNA genes reveals a distinct cluster with a high bootstrap value (86) that encompasses two *trnL* genes of *Agelas nakamurai* (Figure 4a). Conversely, the *trnL* genes of *Ectyoplasia ferox*, *Geodia neptuni*, *Halichondria okadai*, and *Axinella corrugata* are assigned to separate branches. To further investigate this phenomenon, we conducted a comparative analysis of the sequence and secondary structure similarity between the two *trnL* of *Axinella corrugata* and *Agelas nakamurai* (Figure 4b). In the case of *Axinella corrugata*, the two *trnL* sequences share only 59.3% similarity. Additionally, *trnL_1_(uaa)* exhibits an additional loop structure at 25 bp to 39 bp when compared to *trnL_2_(uag)*. On the other hand, despite having different anticodons, the sequence similarity between the *trnL_1_(uaa)* and *trnL_2_(uag)* in *Agelas nakamurai* is as high as 87.5%. Furthermore, both genes share a high degree of similarity in their secondary structure. Meanwhile, it is worth noting that *Agelas schmidti* also exhibits a high similarity between *trnL_1_(uaa)* and *trnL_2_(uag)*, with the sequence similarity of the two *trnL* genes reaching 90.3%.

### 2.6. Nucleotide Diversity and Positive Selection Site Analysis

A sliding window analysis was conducted to identify nucleotide polymorphisms (π) in 14 PCGs among the 17 mitogenomes of Demospongiae sponges (Appendix A). The result reveals significant variations in nucleotide polymorphism across different PCGs, with relatively high levels observed in *atp8*, *nad4L*, and *nad6* (0.284, 0.279, and 0.279, respectively), while *atp9* showed the lowest value of π at only 0.178. The remaining genes exhibit π values ranging from 0.201 to 0.256 (Figure 5a).

To investigate the occurrence of adaptive evolution with lineage specificity in *Agelas* sponges, we performed a positive selection site analysis on 14 PCGs within their mitogenomes. Two species from the *Agelas* were chosen as the foreground branches. The analysis results reveal the presence of a positive selection site in both the *nad3* and *nad5* mitochondrial genomes of the two *Agelas* sponges. Specifically, position 4 in the *nad3* gene and position 439 in the *nad5* gene were found to have positively selected amino acids.

## 3. Discussion

We sequenced the mitochondrial genome of a sponge collected from the South China Sea and identified it as *Agelas nakamurai* based on the *cox1* sequence. The mitochondrial genome of *Agelas nakamurai* is 20,885 bp long with a GC content of 37.6%, which is similar to that of other Demospongiae sponges. We conducted a comparison and found that the mitochondrial genome of *Agelas nakamurai* is 525 bp longer than that of its congener, *Agelas schmidti*. The variation in genome size is primarily due to the expansion of the intergenic region, a characteristic shared with most other Demospongiae sponges [8].

In mitochondrial genome PCG-based phylogenetic analysis, the vast majority of Demospongiae sponges effectively differentiated at the superfamily level. Previous molecular research has primarily classified Demospongiae sponges into five major clades: G0 (Homosclerophorida), G1 (Dictyoceratida, Dendroceratida), G2 (Chondrosida, Halisarcida, and Verongida), G3 (Marine Haplosclerida), and G4 (all other groups) [20,21]. However, the clustering of Dictyoceratida and Dendroceratida within G1 was not readily apparent in our phylogenetic analysis. This discrepancy can be attributed to our use of the Dictyoceratida sponge as the outgroup, which suffered significant loss of mitochondrial PCGs. Nevertheless, the clustering results for the other clades closely matched those reported by Lavrov [20], with *Agelas* sponges positioned within the G4 group of the clade system. The clustering of *Agelas nakamurai* and *Agelas schmidti* undoubtedly indicates their close affinity. Interestingly, the two species of *Agelas* sponges, as well as *Axinella corrugata*, form a monophyletic group and are considered sister groups to other sponges within the *Axinella* superfamily, despite *Axinella corrugata*’s classification as belonging to the *Axinella* superfamily. This finding is consistent with other studies on the evolutionary relationships of Demospongiae based on mitochondrial genomes [19,20].

In terms of gene arrangement, *Agelas nakamurai* is consistent with *Agelas schmidti* of the same genus, suggesting that the gene arrangement of mitochondrial genomes within the genus *Agelas* may be conserved. At the same time, we noticed that *Agelas nakamurai* and *Agelas schmidti* both have a *trnD* deletion. Due to the presence of superwobble phenomena and tRNA import mechanisms in eukaryotes, the lack of the *trnD* gene in *Agelas nakamurai* does not lead to changes in codon usage bias of mitochondrial PCGs (Appendix A). However, it is possible that the lack of *trnD* has implications for the nuclear genome. Because changes in mitochondrial tRNA are often associated with alterations in the nuclear genome. Specifically, the loss of mitochondrial tRNA (mt-tRNA) genes can result in the redundancy of specific translation components in mitochondria, ultimately leading to the loss of corresponding nuclear genes [15]. A notable example is the correlation observed in other non-bilaterian animals, where the loss of mt-tRNA is connected to the loss of nuclear-encoded mitochondrial aminoacyl-tRNA synthetases [22,23]. Future research on the *Agelas* nuclear genome might reveal similar phenomena.

Phylogenetic analysis of tRNA genes in *Agelas nakamurai* sponges reveals a notable deviation from other sponge species in the phylogeny of two *trnL* genes. A comparison indicates that these *trnL* genes in *Agelas nakamurai* exhibit a distinctively high similarity in both sequence and secondary structure of their products. The results indicated that: (i) the evolutionary processes of *trnL* genes in the mitogenome of *Agelas nakamurai* differed comparatively from other sponge species, displaying relative independence; (ii) the duplication and remodeling of one single *trnL* gene resulted in the presence of two *trnL* genes in the *Agelas nakamurai* mitogenome. Similar gene duplication and remodeling events have been observed in the evolution of mitochondrial genes encoding isoacceptor tRNAs [24,25]. Notably, comparable phenomena have been observed in *Agelas schmidti*, supporting the notion of *trnL* gene duplication and remodeling as an evolutionary characteristic of the *Agelas*.

Calculation of nucleotide polymorphism revealed that the *atp8* gene exhibits the highest level of polymorphism in the mitochondrial genome of *Agelas nakamurai* sponge, followed by *nad6*. Conversely, the *atp9* gene exhibits the lowest level of nucleotide polymorphism. In a study conducted by Wang et al. on mitochondrial genomes of various demosponges, it was observed that the *atp8* gene displays the lowest level of conservation in demosponges mitochondria, followed by the *nad6* gene. On the other hand, the *atp9* gene demonstrates the highest level of conservation [13]. These findings suggest that the polymorphic characteristics of the PCGs in *Agelas nakamurai* sponge align with those of the majority of demosponge sponges.

Evolutionary analysis reveals a positive selection site within the *nad3* and *nad5* genes of *Agelas* sponges. Recent studies have unveiled the adaptive evolution of mitochondrial PCGs, which play a crucial role in oxygen utilization and energy metabolism [26]. The *nad3* and *nad5* genes specifically encode crucial subunits of complex I in the mitochondrial electron transport chain [27]. Complex I is responsible for the crucial role of being the primary entry point for the electron transport chain in cellular respiration [28]. As a result, subunits of complex I are highly conserved among different organisms [29]. The presence of positively selected sites within the *nad3* and *nad5* genes implies that these genes have undergone adaptive changes in response to both environmental stress and natural selection. The adaptive variation of these genes may be the key to the survival of *Agelas* sponges in their ecological niche. Although we are unable to determine the biological effects of the two amino acid substitution sites in the *Agelas*, this information still provides us with valuable insights into their evolutionary history.

## 4. Materials and Methods

### 4.1. Sample Collection and DNA Extraction

*Agelas* sp. specimens were collected from the Quanfu Island of Xisha Island from the South China sea. Total genomic DNA was extracted with the TIANamp Marine animals DNA Kit (Tiangen Biotech, Beijing, China) following the manufacturer’s protocol.

### 4.2. Illumina Library Preparation and Sequencing

Paired-end libraries were prepared according to the instructions of the VAHTS Universal DNA Library Prep Kit for Illumina V3 (cat: ND607-02). The size-selected, adapter-modified DNA fragments were PCR-amplified using PE PCR primers and following protocol: polymerase activation (98 °C for 2 min), followed by 10 cycles (denaturation at 98 °C for 30 s, annealing at 65 °C for 30 s, and extension at 72 °C for 60 s) with a final, 4 min extension at 72 °C. DNA libraries were purified by magnetic beads, quantified by RT-PCR. The sequencing process was carried out at the Jiangsu Recbio Technology Co., Ltd. (Taizhou, China), with 150 bp paired end reads.

### 4.3. Sequence Assembly and Annotation

Illumina paired-end reads were filtered according to sequencing quality and trimmed for low-quality bases (quality < 20, p_error_ > 0.01) upstream and downstream. The Chloroplast & Mitochondrial Assembly (CMA) V1.1.1 software (Guangzhou SCGene Co., Ltd., Guangzhou, China) was used to analyze and assemble the raw reads data. Genome confirmation was performed by mapping the paired-end reads to the genome with 100% coverage and insert-size in accordance with the characteristics of the sequencing library. Additional criteria for confirmation included sequencing depth, coverage, and the relationship between the paired end reads. Coding genes were annotated with BLASTX [30], while tRNAs were annotated using tRNAscan-SE v2.0 (web server) [31] and MITOS (http://mitos.bioinf.uni-leipzig.de/index.py, accessed on 12 March 2023) [32]. The predicted boundaries of the rRNA genes were identified by aligning them to rRNA sequences obtained from published sources. These alignments were then manually verified to ensure accuracy.

### 4.4. Species Identification Based on the cox1 Gene

The DNA sequence of the *cox1* gene was extracted and uploaded to both the National Center for Biotechnology Information (NCBI) nucleic acid database [33] and the BOLD Identification System (IDS) [34] for sequence comparison. Based on the identification using the IDS (Database: Species Level Barcode Records, Current: 11 July 2023), the *cox1* barcoding sequence of *Agelas* sp. showed a 100% similarity (query coverage: 63.3%) only to *Agelas nakamurai*, leading us to identify *Agelas* sp. as *Agelas nakamurai*. To further validate the accuracy of the identification result, we performed a phylogenetic analysis using the *cox1* gene. The *cox1* sequences of the 30 species that exhibited the greatest similarity were chosen for the construction of a phylogenetic tree with *Agelas* sp. This selection comprised 5 individuals from the NCBI database and 25 individuals from the IDS database. In total, 9 species from the *Agelas* and 2 species from the *Axinella* genus were included. The phylogenetic tree was constructed using the ML method, with the bootstrap test repeated 1000 times. This analysis was performed using MEGA v11 [35].

### 4.5. Phylogenetic and Gene Arrangement Analysis

To investigate the mitochondrial genomic characteristics of the genus *Agelas*, we obtained the complete mitogenomes of 37 species from the class Demospongiae through the NCBI database. These genomes were then utilized to build a multigene phylogenetic tree, in which *Agelas nakamurai* was included. For this purpose, PhyloSuite v1.2.3 [36] was employed to extract the PCGs, tRNA, and rRNA sequences from the mitogenomes [37]. Once aligned, the extracted sequences were trimmed using Gblocks v0.91b [38] and concatenated using PhyloSuite v1.2.3. The best partition scheme and model for BI and ML were discovered using ModelFinder [39] with the Bayesian Information Criterion (BIC). The results can be found in Appendix A. BI tree was reconstructed using MrBayes v3.27a [40] employing the Markov Chain Monte Carlo method (Generations: 2,000,000; Sampling Freq: 1000) to calculate the posterior probability. Meanwhile, ML trees of 14 PCGs were constructed using IQtree v2.2.2.6 [41], with Ultrafast bootstrap analysis [42] method repeated 1000 times. The mitogenomes of *Agelas nakamurai*, *Agelas schmidti*, and *Axinella corrugata* were visualized by OGDRAW (https://chlorobox.mpimp-golm.mpg.de/OGDraw.html, accessed on 24 September 2022) [43]. The statistics of aspartic acid codons and the calculation of RSCU values were completed using MEGA v11. Data visualization was completed using the ggplot2 package in R v4.3.1 [44].

### 4.6. Analysis of tRNA Duplication, and Remoulding

*Ectyoplasia ferox* (Accession: EU237480.1), *Geodia neptuni* (Accession: AY320032.1), *Halichondria okadai* (Accession: NC_037391.1), *Agelas schmidti* (Accession: NC_010213.1), and *Axinella corrugata* (Accession: AY791693.1) were selected to construct a phylogenetic tree (Neighbor-joining method based on p-distances) of mitochondrial tRNA genes. All tRNA genes undergo verification using tRNAscan-SE v2.0 (web server) and MITOS to ensure precise annotation. The results indicated that the tRNA annotation outcomes from both tools were consistent, although discrepancies were found compared to those on GenBank. Given the potential for GenBank’s annotation results to be derived from an outdated tRNA annotation model, which may have lower accuracy, we opted to utilize the annotation outputs from tRNAscan-SE v2.0 (web server) and MITOS for further analysis. The structure prediction of tRNA was conducted using tRNAscan-SE v2.0 (web server) and visualized with Forna (http://rna.tbi.univie.ac.at/forna, accessed on 19 March 2023) [45].

### 4.7. Nucleotide Diversity and Selection Pressure Analysis

Seventeen mitochondrial genomes of Demospongiae sponges, comprising all 14 PCGs, were chosen for evolutionary analysis. We utilized the sliding window mode of DnaSP v6 (window length: 200, step size: 25) [46] to calculate the nucleotide diversity of 17 Demospongiae sponge mitochondria. The phylogenetic tree necessary for the analysis of selection pressure was constructed using the ML method, utilizing the sequences of 14 PCGs. The construction process closely followed the methodology outlined in the previous section titled “Phylogenetic and Gene Arrangement Analysis”. The branch-site model was utilized to detect amino acid sites under positive selection in the lineage of the *Agelas*. The identification of positive selection at a site can only be confirmed if the analysis results of Naive Empirical Bayes (NEB) or Bayes Empirical Bayes (BEB) indicate a probability higher than 0.995.

## 5. Conclusions

In this study, we conducted a comprehensive analysis of the mitogenome of *Agelas nakamurai*. The comparative analysis revealed that *Agelas nakamurai* and *Agelas schmidti* share identical gene arrangement in terms of PCGs, rRNA genes, and tRNA genes. Additionally, we have identified a deletion in the *trnD* gene and replication and remodeling of the *trnL* gene in the mitogenome of *Agelas nakamurai*. Furthermore, our evolutionary analysis also indicates the presence of lineage-specific positive selection sites on the *nad3* and *nad5* genes in *Agelas*.

## Figures and Tables

**Figure 1 ijms-25-00357-f001:**
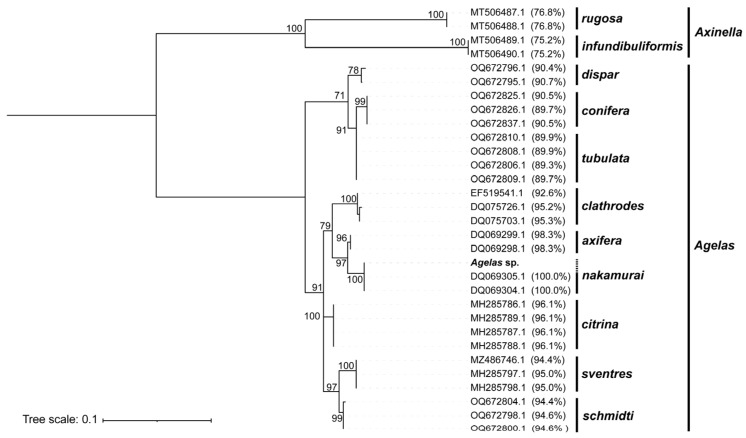
ML tree depicting the *cox1* genes among *Agelas* sp. and other sponges from the genera *Agelas* and *Axinella*. The accession numbers of the *cox1* genes and their sequence similarity (query coverage: 56.3%~63.1%) with the *cox1* of *Agelas* sp. are indicated at the end of each leaf. Percent bootstrap values are shown near the nodes. Only values above 70 are displayed.

**Figure 2 ijms-25-00357-f002:**
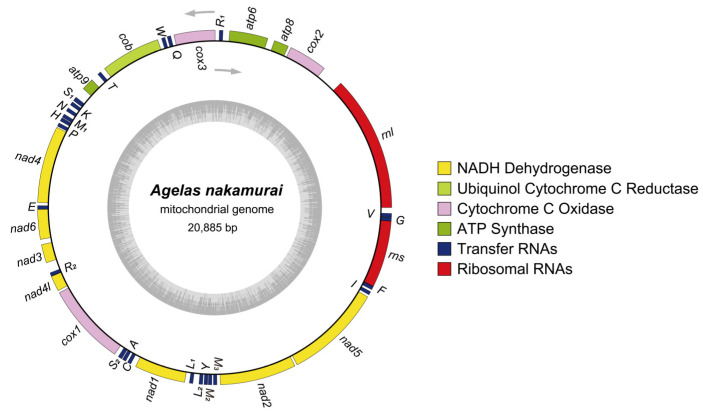
Mitogenome maps of *Agelas nakamurai*. The gray arrows represent the direction of transcription. Distinct colors correspond to various genes. The tRNA genes are denoted by single-letter abbreviations representing the accepted amino acid. The gray inner ring represents the GC content of the sequence.

**Figure 3 ijms-25-00357-f003:**
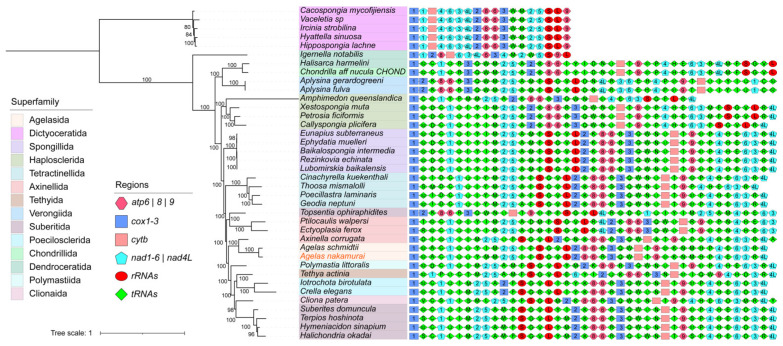
BI phylogenetic tree illustrating the mitogenomes of 38 species from the Demospongiae class of sponges. *Hippospongia lachne*, *Hyattella sinuosa*, *Ircinia strobilina*, *Vaceletia* sp., and *Cacospongia mycofijiensis* were used as outgroup for tree construction. The numbers adjacent to the nodes indicate the percentage of posterior probability values, with only values exceeding 70 being shown. Leaf labels are color-coded to represent different superfamilies. On the right side, a gene arrangement diagrams show the arrangement of mitochondrial genes in 38 Demospongiae sponges, with genes represented by various colors and shapes. This plot exclusively depicts the relative positioning of genes on the mitochondria while disregarding information related to the length. To facilitate illustration, the PCGs from same family are abbreviated in Arabic numerals, for example: the blue rectangle in the figure represents the genes of the *cox* family, and if the number “1” appears in the rectangle, it represents the gene locus *cox1*, and so on. tRNA genes are abbreviated by their corresponding ligand amino acids. rRNA includes two types, *rnL* and *rnS*, abbreviated as “L” and “S” respectively.

**Figure 4 ijms-25-00357-f004:**
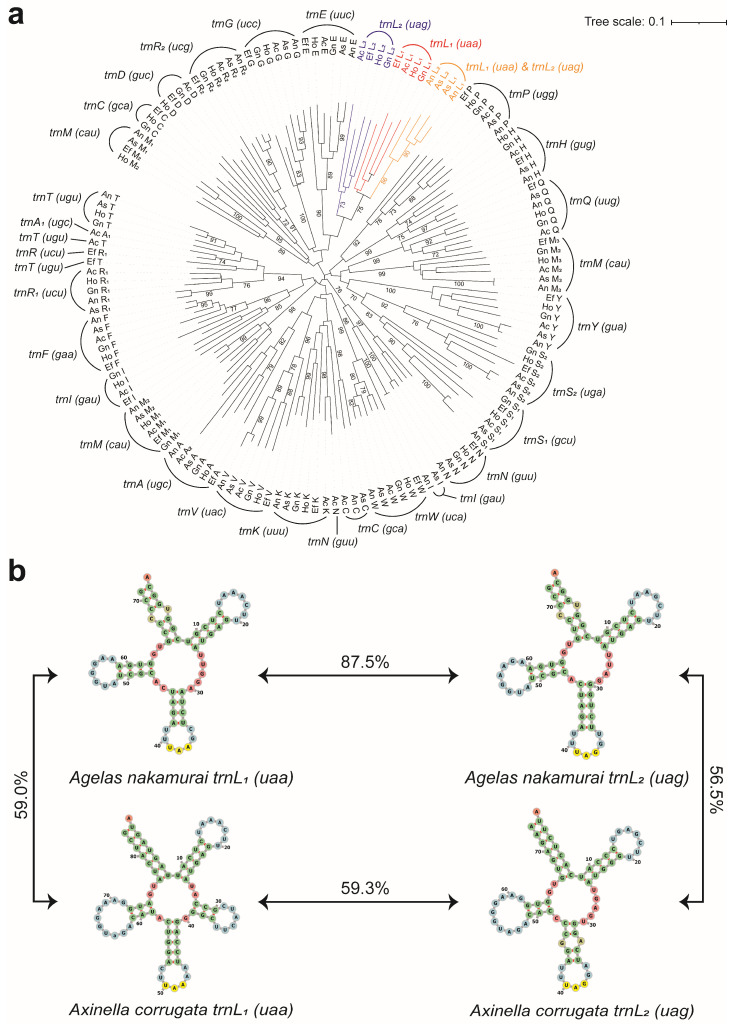
Evolutionary analysis of the tRNA genes of *Agelas nakamurai*. (**a**) An illustrated neighbor-joining tree displaying the relationships between mitochondrial tRNA genes from *Agelas nakamurai* (An), *Agelas schmidti* (As), *Axinella corrugata* (Ac), *Ectyoplasia ferox* (Ef), *Geodia neptuni* (Gn), and *Halichondria okadai* (Ho). Bootstrapped values are indicated on the branches of the tree. Only values above 70 are displayed. Colored branches and fonts are used to indicate the *trnL* genes from different evolutionary branches. (**b**) Predicted secondary structure of *Agelas nakamurai trnL*_1_*(uaa)*, *Agelas nakamurai trnL*_2_*(uag)*, *Axinella corrugata trnL*_1_*(uaa)*, and *Axinella corrugata trnL*_2_*(uag)*. The sequence similarity between tRNA genes is indicated in the middle of the arrows.

**Figure 5 ijms-25-00357-f005:**
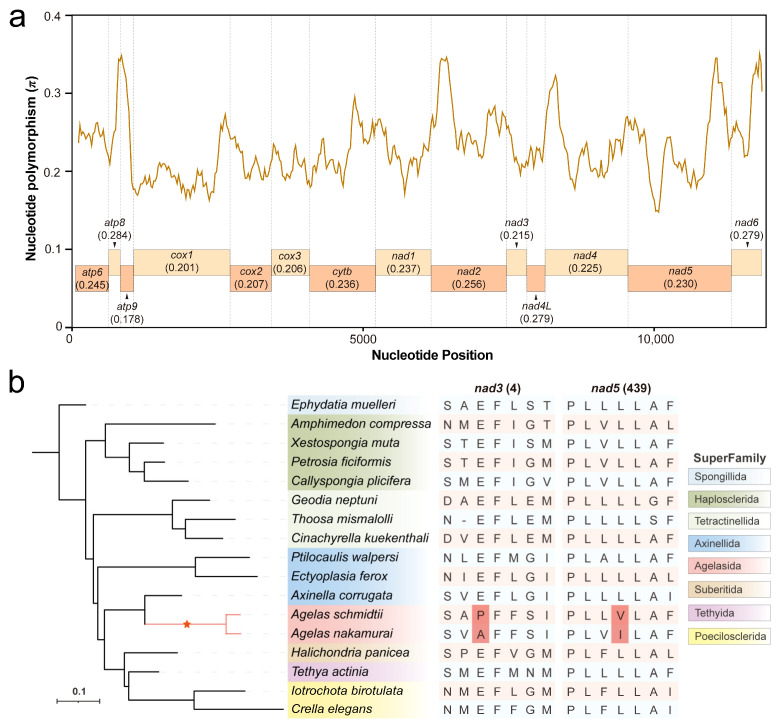
Nucleotide diversity and evolutionary analysis of 14 PCGs in the *Agelas*. (**a**) A sliding window analysis of PCGs of *Agelas nakamurai*. The curve represents the level of nucleotide polymorphisms (π), while the rectangle below illustrates the specific PCGs and their corresponding average π values. (**b**) Analysis of positive selection sites with *Agelas* as the foreground branch. The *Agelas* genus is indicated with an asterisk on the phylogenetic tree. Amino acids residing in the positively selected sites are distinguished by a red background.

## Data Availability

The genome sequence data that support the findings of this study are openly available in GenBank of NCBI at https://www.ncbi.nlm.nih.gov (accessed on 2 February 2023) under accession no. OQ363829.

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
