# Peer review of "Characterization of the Complete Mitochondrial Genome of Agelas nakamurai from the South China Sea"

_ijms, 2023, doi:10.3390/ijms25010357_

Round 1

Reviewer 1 Report

Comments and Suggestions for Authors

General comments

The manuscript by Lu et al. describes the mitogenome sequence of Agelas nakamurai, a member of the Demospongiae sponges, and compares it to that of other sponges. As a mitogenome paper, this one is fairly straightforward, but the presentation can be improved substantially and not all of the results are fully reliable.

Specific comments

1.    The “Results and Discussion” section begins directly with “2.1. Species identification of Agelas sp.”, directly launching into phylogenetic analysis. Which is really out of place – the natural organization of such a paper is to first describe how the specimens were collected, sequenced, and assembled. But that is completely missing from the results section in the current version of the manuscript. That needs to be fixed.

2.    Figure 2 is misleading in its current form because the comparison of gene presence and synteny shows both protein coding genes and tRNAs as geometric forms of the same size. It should be reworked, by showing the full genomes in their actual length, with protein coding genes and tRNAs show in their true length too.

3.    Figure 3, which is the first one to show the Agelas nakamurai mitogenome, naturally comes before such a Figure 3 in the logical order of the results.

4.    Line 91 says that “Like other sponges, the mitogenome of Agelas nakamurai exhibits a highly compact structure”. But that is not a common property of sponges, it is a common property of all metazoans.

5.    The extended discussion about the absence of trnD (Lines 146-162) is really superfluous. Not all mitogenomes encode all tRNAs, and often that is simply because tRNAs are imported from the nucleus. As the authors themselves mention at the end.

6.    The whole dN/dS analysis is based on very little data and does not produce any really reliable or notable results. I would remove it, as it is neither necessary for the paper nor does it add anything important to it.

7.    There are quite a few typos and other errors to be fixed. Just a few examples from the beginning:

(a)    Line 47: “Sponge possess a more diverse mitogenome compared with bilaterian animals” should be “Sponges possess a more diverse mitogenome compared with bilaterian animals”

(b)    Line 49: “tRNA genes ,the presence of repetitive elements with” should be “tRNA genes, the presence of repetitive elements with”

(c)     Line 55: “from the Agelas, has had its mitogenome” should be “from the Agelas genus, has had its mitogenome”

(d)    Line 58: “Here, we presented the first report on the complete mitogenome of an Agelas sp.” should be “Here, we present the first report on the complete mitogenome of an Agelas sp.” (e) etc.

Comments on the Quality of English Language

There are quite a few typos and errors that need to be fixed, but overall it is not too bad.

Author Response

We would like to express our gratitude for your professional review of our articles. As per your concerns, there are several issues that require attention. We have carefully considered your suggestions and have made extensive revisions to the previous draft. Additionally, based on the feedback from another reviewer, we have restructured the article, splitting the "Results and Discussion" into two separate parts. We understand that this may cause inconvenience for your review, and we apologize for any inconvenience this may cause. Detailed information on the changes can be found in the attachment.

Thank you once again for your valuable comments on the revision of this manuscript.

Reviewer 2 Report

Comments and Suggestions for Authors

IJMS- 2723548 review report

The topic of the manuscript is of interest. The authors performed a comprehensive analysis to come up with significant conclusions. However, some improvements are required in the manuscript.

Major revisions,

1.     The results and discussion could be in separate sections.

2.     It is not clear if authors have identified Agelas nakamurai and compared it with Agelas schmidti in the study, if they identified it in the Agelas specimens, the authors could consider giving a name of the investigated genome and show compassion genomes with Agelas schmidti.

3.     Page 2, lines 54-55, it says “only one species, Agelas schmidti (Accession: NC_010213.1) [17] from the Agelas has had its mitogenome 55 sequenced.” However, some other studies have reported sequencing of some sponge families (see PMID: 33473527, PMID: 18628961). Even the cited paper in this study (https://doi.org/10.1016/j.ympev.2008.05.014) has reported analysis of several demosponge mitochondrial genomes. Please clarify it.

4.     Figure 3, page 4, should illustrate the identified genome with the given name.

5.     Page 10, lines 349-350, it says “genome sequence data that support the findings of this study are openly available in GenBank under accession no. OQ363829”. The records are not accessible at GenBank under accession no. OQ363829.

6.     Page 10, lines 345-346, it says “All animal experiments were conducted”. It is not clear which animal experiments were conducted since the research in the manuscript has been conducted for comprehensive analysis of the mitogenome of Agelas nakamurai.

7.     Lines 327 to 328, it says “However, to confirm the universality of these characteristics within the genus Agelas, it is necessary to verify the complete mitogenome of Agelas sponges from multiple species.” This statement shows that the study is incomplete, please clarify it.

Minor revision,

Introduction-

1.     Line 36 to 40, the statements need more references, the current reference (https://doi.org/10.1016/j.toxicon.2004.04.001) is not sufficient to cite the information.

Results and Discussion-

1.     Lines 71, 72, it says “Importantly, the individual we investigated clusters together with two other Agelas nakamurai specimens”. Add the accession number of “Agelas nakamurai specimens” which clustered with Agelas species. Also add the name of the “Agelas species” that is being investigated.

Materials and Methods-

1.     Page 8, lines 249-250 it says “Illumina DNA, libraries were prepared”. It could be improved by providing more information about which library kits and processes were utilized.

2.     Page 8, line 254-255, a reference or a URL of the software “Chloroplast & Mitochondrial Assembly (CMA) V1.1.1” need to be provided.

3.     Page 8, Line 257, the reference for “tRNAscan-SE 2.0” should be “DOI: 10.1093/nar/25.5.955” and the reference for “MITOS” DOI: 10.1016/j.ympev.2012.08.023.

4.     Provide the reference for blastx line 256.

5.     Line 259 a reference to the “National Center for Biotechnology Information (NCBI) nucleic acid database” needs to be provided.

6.     Line 263, it says “100% similarity”, please also provide the query coverage of this analysis.

7.     Line 275, “NCBI database” a reference is required.

8.     Line 278, the reference needs to be improved, it should be “DOI: 10.1093/oxfordjournals.molbev.a026334”.

9.     Why have there been two different versions of MEGA line 271 and line 287?

10.  Which R package has been used, line 288?

11.  Please check all the citations carefully.

12.  Which tool has been used to create the circular figures (figure 3; page 4) of the Mitogenome of identified genomes of Agelas nakamurai.

Author Response

We feel great thanks for your professional review work on our article. As you are concerned, there are several problems that need to be addressed. According to your nice suggestions, we have made extensive corrections to our previous draft, the detailed corrections are listed in the attachment.

Once again, we are grateful for your insightful comments and constructive criticism. We look forward to hearing your thoughts on the revised manuscript.

Round 2

Reviewer 1 Report

Comments and Suggestions for Authors

The authors have mostly answered my comments and suggestions and I don't think there is a much of a point in persisting with the remaining outstanding issues

Comments on the Quality of English Language

Could and should be improved, but there is really no point debating this further.

Author Response

Thank you again for your positive comments and valuable suggestions to improve the quality of our manuscript. We have tried our best to polish the language in the revised manuscript. And here we did not list the changes but marked in yellow/grey in the revised paper. Yellow represents added content, and gray represents language corrections. We appreciate for Editors/Reviewers’ warm work earnestly and hope that the correction will meet with approval.

Reviewer 2 Report

Comments and Suggestions for Authors

Author Response

Thank you again for your positive comments and valuable suggestions to improve the quality of our manuscript. We have highlighted the changes in yellow/grey in the revised paper. Yellow represents added content, and gray represents language corrections. Detailed modification content please see the attachment. We appreciate for Editors/Reviewers’ warm work earnestly and hope that the correction will meet with approval.
